# Direct Formation of ZIF-8 Crystal Thin Films on the Surface of a Zinc Ion-Doped Polymer Substrate

Takaaki Tsuruoka *, Kaito Araki, Kouga Kawauchi, Yohei Takashima and Kensuke Akamatsu

Department of Nanobiochemistry, Frontiers of Innovative Research in Science and Technology (FIRST), Konan University, 7-1-20 Minatojimaminami, Chuo-ku, Kobe 650-0047, Japan
* Correspondence: tsuruoka@konan-u.ac.jp

**Abstract:** Thin films of metal–organic frameworks (MOFs) on polymer substrates and MOF/polymer mixed-matrix membranes play crucial roles in advancing the field of gas separation membranes. In this paper, we present a novel method for the direct formation of continuous ZIF-8 crystal films on a polymer substrate doped with $Zn^{2+}$. Our approach involves ion exchange between the doped zinc ions within the substrate and sodium ions in the presence of a $CH_3COONa$ additive, as well as interfacial complexation with eluted zinc ions and 2-methylimidazole (2-MeIM). The key factors affecting the formation of ZIF-8 crystals on the substrate were the concentrations of $CH_3COONa$ and 2-MeIM. A time-course analysis revealed that the nucleation rate during the early stages of the reaction significantly affected the surface morphology of the resulting ZIF-8 crystal films. Specifically, a higher nucleation rate led to the formation of continuous small ZIF-8 crystal films. This innovative approach enables the fabrication of densely packed, uniform ZIF-8 crystal films.

**Keywords:** metal–organic frameworks (MOFs); interfacial reaction; ion exchange reaction

## 1. Introduction

Metal–organic frameworks (MOFs), a class of highly ordered porous materials, have been extensively studied because of their unique properties [1–3]. These properties are associated with interior pore structures, which can be controlled by combining them with nodes consisting of metal ions and organic ligands [4,5]. Therefore, MOFs are promising materials for applications in gas storage [6–8], separation [9–11], catalysis [12–14], and sensing [15–17]. Over the past decade, numerous studies have focused on the design, synthesis, characterization, and utilization of bulk MOF crystals; however, certain applications of MOFs, such as gas separation using membranes, require control of the nucleation and growth of the MOF crystals on a substrate, which is important for constructing MOF films with specific desired properties [18–22].

In recent years, the preparation of MOF membranes has garnered significant attention because of their unique properties derived from the efficient use of pore space and integration of specific functions [23–27]. To achieve this, sequential processes involving uniform nucleation on a substrate and subsequent nuclear growth must be employed. These processes are essential for fabricating continuous MOF films without cracks or inter-crystalline gaps. However, the equilibrium reaction for MOF crystal formation leads to preferential crystal growth, making the preparation of continuous MOF crystal films using thermodynamic synthetic approaches challenging. Consequently, the development of a universal and straightforward approach to control nucleation and growth processes on a substrate could provide deeper insights into the growth mechanism of framework materials, and pave the way for constructing high-performance MOF films.

Various methods have been developed for fabricating thin films composed of MOF crystals. These methods involve a straightforward layer-by-layer deposition process [28–30] and direct formation on functionalized substrates [31–34]. We also demonstrated a simple interfacial approach for MOF-crystal-based films using a metal-ion-doped polymer

substrate [35–37]. For example, in the case of an MOF constituting carboxylate ligands, this method relies on an ion-exchange reaction between metal ions doped into polyimide substrates and the protons of organic ligands containing carboxylic acid groups. In contrast, in the case of an MOF constituting amine ligands, the addition of an ion-exchange promoter such as sodium nitrate facilitates the ion-exchange reaction of metal ions doped in the polymer substrate, leading to interfacial complexation between the eluted metal ions and organic ligands without carboxylic groups [38]. This approach offers a straightforward and rational process for fabricating amine-based MOF films. However, controlling the elution rate of the doped metal ions in the polymer substrate is difficult because the dissociation rate of the additives (ion-exchange promoters) is too high to control the ion-exchange reaction between the doped metal ions and dissociated sodium ions. Therefore, a fast ion-exchange reaction induces a nucleation-preferential process in the initial stage of the reaction. In order to induce crystal growth within this approach, it is essential to utilize a significant volume (500–1000 mL) of reaction solution containing the eluted metal ions (a dilute $Zn^{2+}$ solution wherein crystal growth can be induced). The reaction time required to complete the crystal growth is 72 h, which is longer compared to the MOF formation approach using a metal-ion-doped polymer substrate. This limits the application of this synthetic approach for producing continuous MOF films.

Herein, we propose a simple strategy for depositing continuous amine-based MOF (ZIF-8) crystal films on metal-ion-doped polymer substrates using basic salts, such as sodium acetate, instead of neutral sodium nitrate. A sodium acetate solution induces an ion-exchange reaction between the doped metal ions in the polymer substrate and the sodium ions, resulting in the elution of the doped metal ions from the substrate. After this ion-exchange reaction, all the eluted metal ions are consumed on the surface of the substrate, either via complexation with organic ligands or the formation of zinc hydroxide. Subsequently, zinc hydroxide is converted to ZIF-8 in the presence of excess organic ligands via a replication reaction [39–42]. This methodology allows understanding of the basic principles of MOF crystal formation, and provides important insight into the fabrication of continuous MOF crystal films.

## 2. Results and Discussion

Preparation of ZIF-8 crystal films on a $Zn^{2+}$-doped polymer substrate using the $CH_3COONa$ additive: to form a ZIF-8 crystal on the substrate, $Zn^{2+}$-doped polyimide films were immersed in a solution (methanol/$H_2O$ = 1/2) containing 2-MeIM (100 mM) and $CH_3COONa$ (50 mM) followed by heating at 80 °C for 60 min. The SEM image of the obtained samples (Figure 1A) revealed that rhombic dodecahedral crystals with $396 \pm 99$ nm were densely formed on the substrate. In addition, upon observing the void areas wherein the polyimide was exposed, the crystalline films were not formed by stacked crystals on the polyimide surface, the resulting films consisted of a single-layered crystals. The cross-sectional SEM image also indicated the ZIF-8 crystals were densely packed and connected in the films, which indicates that continuous thin films without the void could be successfully prepared by the present approach (Figure 1B). The MOF thin films were found to pass a scotch tape test (adhesive force > 1.18 N cm$^{-1}$). This may have been caused by the chemical interaction between zinc ions of MOF crystals and carboxylic groups of substrates. XRD measurements of the obtained sample confirmed the formation of ZIF-8 crystals on the polyimide substrate, and a characteristic peak originating from (110) plane of ZIF-8 was observed around 7.3 degrees, and all other peaks with weak intensity were assigned to the simulated ZIF-8 and halo patterns of the polyimide (Figure 1C). Moreover, XPS measurement of the obtained sample showed that the binding energy of $2p_{3/2}$ orbital of Zn was observed at 1022.0 eV, which is consistent with the literature value [43], indicating the formation of ZIF-8 crystals (Figure S1A).

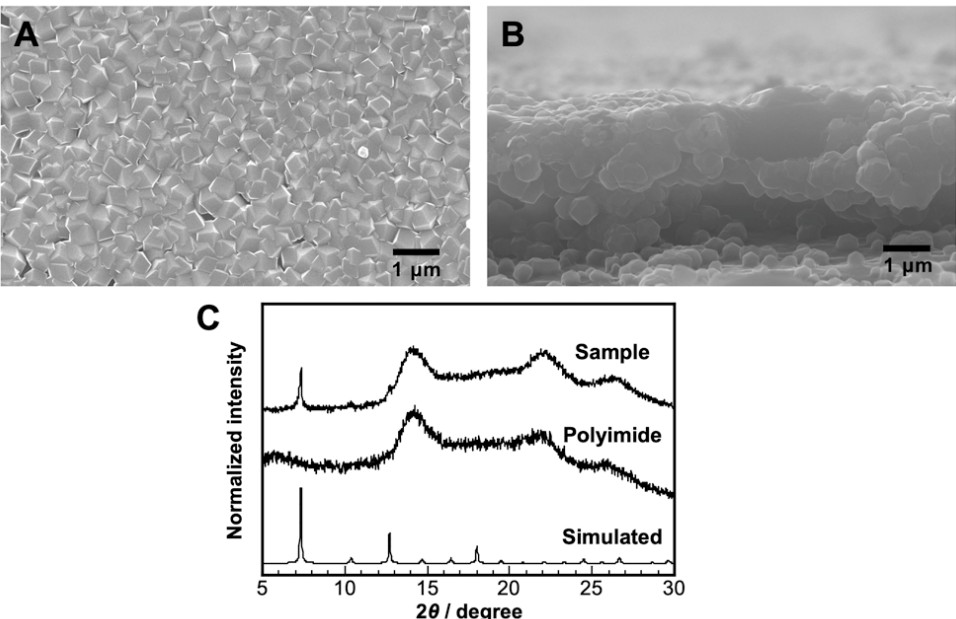

**Figure 1.** SEM images ((**A**): top-view image and (**B**): cross-sectional image) and XRD pattern (**C**) of the obtained sample prepared by the present approach using the reaction solution containing 2-MeIM (100 mM) and $CH_3COONa$ (50 mM).

Our previous approach using a $NaNO_3$ additive led to the rapid elution of doped $Zn^{2+}$ at the initial stage of the reaction, resulting in the formation of a large amount of free $Zn^{2+}$ in the reaction solution. To determine the amount of eluted $Zn^{2+}$ in the reaction solution in the present approach using $CH_3COONa$ additives instead of $NaNO_3$, the amounts of $Zn^{2+}$ and $Na^+$ in the reaction solution and polyimide films before and after the reaction were quantified via ICP measurements. Consequently, $Zn^{2+}$ was not detected in the solution obtained after the reaction. In addition, the amount of $Zn^{2+}$ of the resulting samples after reaction remained unchanged at ca. 1300 $nmol·cm^{-2}$, compared to that of initial $Zn^{2+}$-doped polymer film. In contrast, the amount of $Na^+$ in the polyimide films after the reaction was ca. 2800 $nmol·cm^{-2}$, which is twice the initial $Zn^{2+}$ content of the polyimide films. These results demonstrate that the doped $Zn^{2+}$ in the polyimide films was completely converted to ZIF-8 crystals. These results indicated that the present approach is capable of converting all the doped $Zn^{2+}$ to the desired MOF without leaching unreacted $Zn^{2+}$ into the reaction solution.

Influence of the concentration of the $CH_3COONa$ additive on the size of ZIF-8 crystals: It is crucial to regulate the elution rate of doped $Zn^{2+}$ from the polymer substrate to effectively control the nucleation and growth rate of the formation of ZIF-8 crystals using the present approach. Thus, to evaluate the effect of the $CH_3COONa$ additive on ZIF-8 crystal size, the concentration of $CH_3COONa$ was systematically altered within the range of 0 to 100 mM, whereas the concentration of 2-MeIM and the amount of doped $Zn^{2+}$ were maintained at 100 mM and ca. 1300 $nmol·cm^{-2}$, respectively. Initially, to confirm the effect of $CH_3COONa$ without 2-MeIM on the surface morphology of the $Zn^{2+}$-doped polymer substrate, the substrate was immersed in a solution containing only $CH_3COONa$. Amorphous-like deposits were formed on the substrate (Figure S2A). This result was supported by the XRD measurements, as no distinct peaks corresponding to crystalline materials were observed (Figure S2D). The alkaline properties of the reaction solution containing $CH_3COONa$ may lead to the formation of amorphous zinc hydroxide on the polymer substrate. Therefore, we evaluated the chemical state of the Zn of the resulting sample by XPS measurement. The binding energy of the $2p_{3/2}$ orbital of Zn in the $Zn^{2+}$-doped polyimide substrate was approximately 1022.6 eV, almost the same as that after $CH_3COONa$ treatment (Figure S1A). This result was consistent with previous reports

evaluating the binding energy of zinc hydroxide [44,45], suggesting that the $CH_3COONa$ treatment led to the formation of zinc hydroxide on the polyimide substrate. ZIF-8 was prepared without the $CH_3COONa$ additive. SEM analysis of the resulting sample revealed that nanosized crystals were sparsely distributed on the polymer substrate (Figure S2B). However, no distinct peaks associated with the ZIF-8 framework were observed in the XRD patterns, because the number of ZIF-8 crystals were quite small (Figure S2D). Furthermore, a significant portion of doped $Zn^{2+}$ remained within the polyimide films. These results suggested that the $CH_3COONa$ additives led to ZIF-8 formation on the substrate by $Zn^{2+}$ leaching from the substrate. The reaction was carried out using 25 mM $CH_3COONa$; a larger number of crystals with a size of $167 \pm 28$ nm were formed on the substrate compared to the same without the additive (Figure 2). From the XRD pattern of the resulting sample, a characteristic peak with weak intensity was observed due to the formation of a relatively small number of crystals. With an increase in the $CH_3COONa$ concentration to 100 mM, the crystals formed densely on the substrate and the crystal size increased to $853 \pm 153$ nm. This result was consistent with the XRD measurement result, which showed sharp and intense peaks. However, when the concentration of $CH_3COONa$ increased above 150 mM, a small number of crystals with a size of $1930 \pm 176$ nm were formed on the substrate, and the peak intensity of the XRD pattern became slightly weaker as compared that of the sample prepared using the reaction solution containing $CH_3COONa$ at 100 mM. These results suggest that $CH_3COONa$ plays a pivotal role in influencing the crystal growth rate of ZIF-8. At low concentrations, $CH_3COONa$ accelerates the rate of ZIF-8 nucleation, owing to an increase in the elution rate of $Zn^{2+}$ from the polyimide substrate. Conversely, it is well known that $CH_3COONa$ can act as a modulator in MOF formation; in a solution containing a high concentration of $CH_3COONa$, the $CH_3COONa$ additive not only acts to increase the rate of $Zn^{2+}$ elution, but also acts as a modulator of MOF formation, resulting in a growth-rich process of ZIF-8 formation.

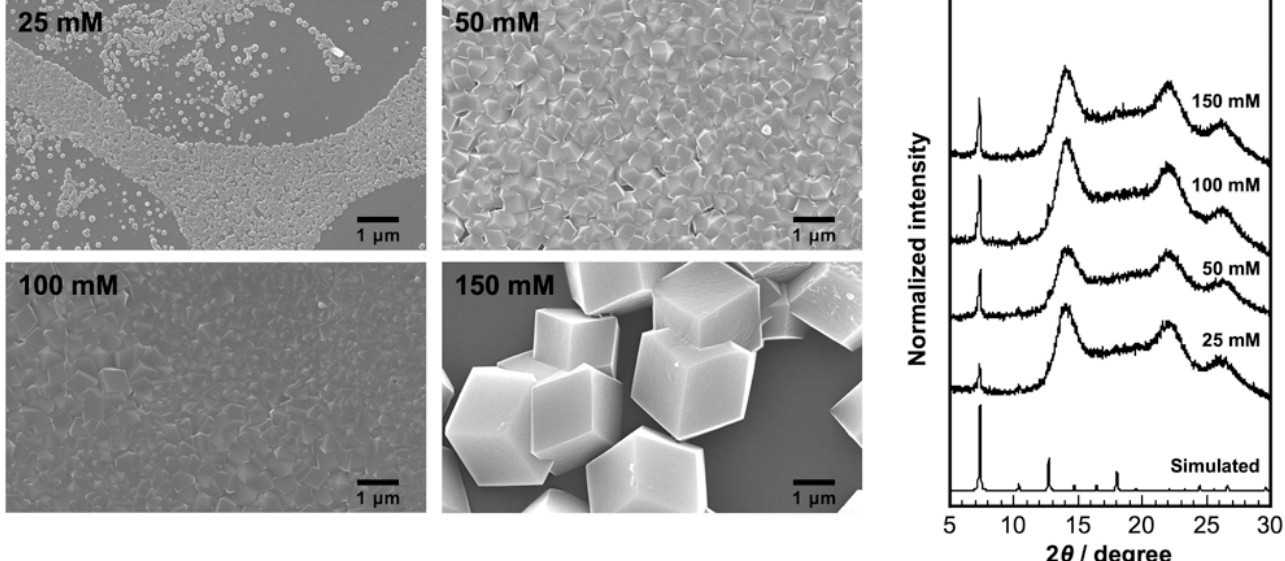

**Figure 2.** SEM images and XRD patterns of the samples prepared by using reaction solutions containing different concentrations of $CH_3COONa$ (concentration of 2-MeIM: 100 mM).

Influence of the concentration of 2-MeIM and the $CH_3COONa$ additive: The influence of the concentrations of the $CH_3COONa$ additive and 2-MeIM on the number and size of the obtained ZIF-8 crystals is summarized in Figure 3. First, focusing on the effect of the $CH_3COONa$ additive concentration, at a ligand concentration of 100 mM, as the additive concentration increased, the crystal size increased without a significant change in the number of crystals. Interestingly, larger crystals are sparsely formed on the substrate at an additive concentration of 150 mM. At a ligand concentration of 200 mM, a similar trend

is observed for additive concentration in the range from 25 to 150 mM. In addition, ZIF-8 crystals are densely formed and interconnected with the substrate at additive concentrations greater than 50 mM. As mentioned above, at low additive concentrations below 25 mM, the present approach leads to a nucleation-rich process. Conversely, at higher additive concentration, the crystal growth of ZIF-8 became a preferential process, resulting in the formation of larger-sized ZIF-8 crystals.

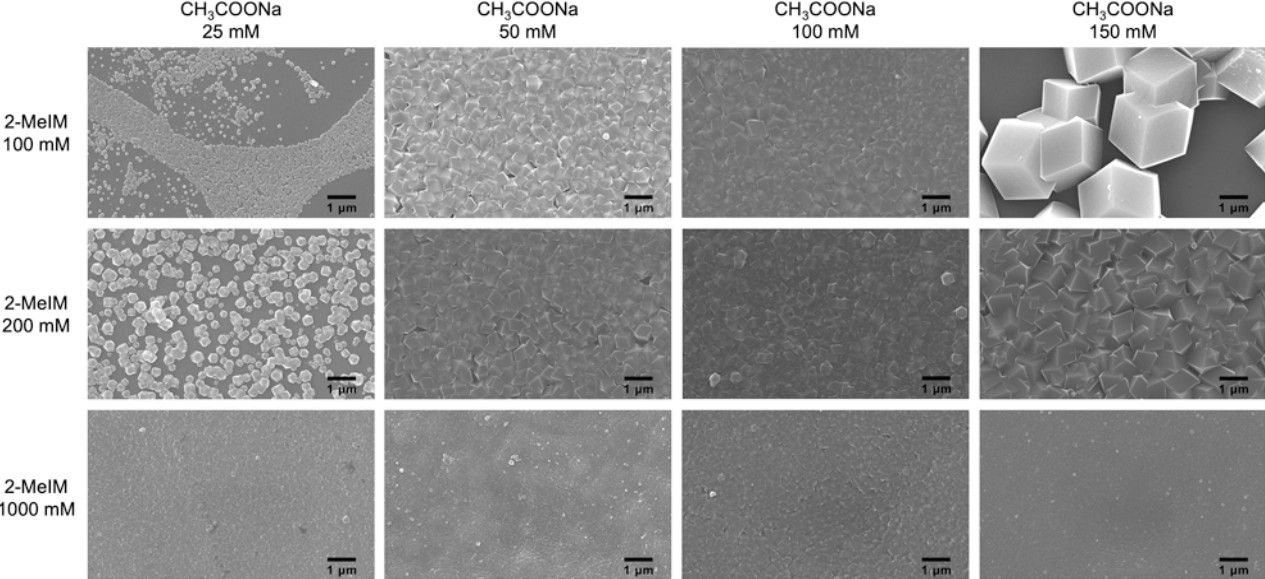

**Figure 3.** SEM images of the crystals prepared using reaction solutions containing different concentrations of 2-MeIM and $CH_3COONa$.

When the ligand concentration was increased up to 1000 mM, the crystal size increased from about 80 (additive concentration at 25 mM) to 180 nm (additive concentration at 150 mM). The impact of additive on crystal growth was less pronounced compared to when the ligand concentration was below 200 mM. This result suggested that the complex formation between $Zn^{2+}$ and organic ligands became significantly faster as the organic ligand concentration increased. Therefore, the nucleation process was accelerated at higher ligand concentrations, meaning that this process is a rate-determining step in the case of excessive organic ligands. Generally, in classical crystal growth following LaMer's law, a higher reactant concentration leads to a nucleation-rich process, which is consistent with previous approaches. Continuous thin films consisting of ZIF-8 crystals with controlled sizes can be prepared using this approach by adjusting the additive and ligand concentration.

Changes in the size and number of crystals during reaction: Time-dependent changes in the surface morphology of the resulting samples prepared with a constant concentration of 2-MeIM (100 mM) are shown in Figures S3–S6, as the concentration of the $CH_3COONa$ additive varied. For example, as shown in Figure S2, at a 25 mM concentration of the $CH_3COONa$ additive, ZIF-8 crystals were formed in the initial stages of the reaction within 1 min. In addition, the number and size of crystals remained almost unchanged during the reaction. It is also noteworthy that only ZIF-8 crystals were observed on the polyimide surface, and there was no detection of amorphous structures such as zinc hydroxide. This result implied that the replication reaction from zinc hydroxide to the ZIF-8 framework was exceptionally fast in this set of experimental conditions. Since the binding energies of the $2p_{3/2}$ orbital of Zn with ZIF-8 crystals and zinc hydroxide are different, XPS measurements were used to confirm the formation of zinc hydroxide in the initial stages of the reaction in the presence of both 2-MeIM and $CH_3COONa$. As shown in Figure 1B, the XPS spectra of the Zn 2p orbital for the resulting sample prepared with different reaction times displayed two peaks at 1022.0 eV and 1044.9 eV, which correspond to the binding energies of Zn

$2p_{3/2}$ and Zn $2p_{1/2}$, respectively. The binding energy of these peaks was almost constant as a function of reaction time, and no other peaks were newly observed. This result was consistent with the SEM observation of the resulting samples in the initial stage of the reaction. On the other hand, the peak intensity became more pronounced as the reaction time increased. It is well known that XPS measurement can only analyze the surface chemical state because the escape depth of photoelectrons is in the order of nanometers. Therefore, the increase in the peak intensity demonstrated that the $Zn^{2+}$ doped inside the polyimide substrate was converted into ZIF-8 crystals on the surface of polyimide, and that the amounts of ZIF-8 on the surface increased with increasing reaction time. In contrast, at concentrations above 50 mM, numerous small ZIF-8 crystals were deposited within a reaction time of 3 min. After a reaction time of 10 min, the number of crystals slightly decreased, and the size of the crystals increased with reaction time. As a result, continuous membranes based on ZIF-8 crystals were formed on the polymer substrate. However, at a higher concentration of 150 mM, a large number of small-sized crystals were observed after the 1 min stage of the reaction. However, with an increase in reaction time, the number of crystals decreased, accompanied by an increase in crystal size, indicating that a higher additive concentration leads to a growth-rich process. A crucial aspect in utilizing the present approach to form ZIF-8 crystals is ensuring an appropriate rate of coordination of the eluted $Zn^{2+}$ from the polymer substrate and 2-MeIM at the polymer surface. To further investigate the impact of ligand and additive concentrations on the size and number of crystals during the reaction, SEM observations of the resulting samples prepared at various concentrations of 2-MeIM and $CH_3COONa$ were carried out as a function of reaction time (Figures S7–S14). When the 2-MeIM and $CH_3COONa$ concentration was 200 and 25 mM, respectively, numerous crystals were observed to be deposited on the polymer after 1 min of reaction time. The number of crystals was almost constant, while the crystal size slightly increased during the reaction, implying that the nucleation was a dominant process under this condition. On the other hand, when the ligand concentration remained at 200 mM and the $CH_3COONa$ concentration increased above 50 mM, the nucleation process appeared to be complete within 10 min, and the number of crystals decreased as the reaction time increased. In contrast, after a reaction time greater than 10 min, the size of crystals gradually increased, indicating that crystal growth preferentially occurred. As shown in Figure 3, the effect of the modulator arising from the $CH_3COONa$ additive appeared to be suppressed at a 2-MeIM concentration of 100 mM. Upon characterization of the variation in the size and number of crystals with reaction time, both the size and number of crystals remained unchanged, suggesting a faster nucleation rate in the early stages of the reaction.

To gain deeper insight into the nucleation and growth of the ZIF-8 crystals, changes in the number and size of the crystals during the reaction were plotted against systematically changing concentrations of the $CH_3COONa$ additive in the reaction solution, whereas the concentration of 2-MeIM was maintained (Figure 4). As shown in Figure 4, at lower 2-MeIM concentrations (100 and 200 mM), numerous crystals were formed in the early stages of the reactions. Subsequently, the size of the crystals increased, and the number of crystals decreased with reaction time. With a few exceptions (2-MeIM concentration at 100 mM and $CH_3COONa$ concentrations at 25 and 150 mM), the area of the bare polyimide surface (intercrystalline gaps) decreased with the increasing reaction time (Figures S4, S5, S8–S10). The fact that the void spaces was reduced without an increase in the number of crystals is thought to be due to the fusion of adjacent crystals as a result of crystal growth during the reaction. These results demonstrated that the present approach involves the nucleation in the initial stage of the reaction, followed by the crystal growth in the middle of the reaction, facilitating the formation of continuous ZIF-8 crystal films. In contrast, at a higher 2-MeIM concentration (1000 mM), both size and number of crystals remained constant regardless of the reaction time. This result indicated that a higher ligand concentration leads to a remarkably face nucleation rate in the initial stage of the reaction, and the nucleation is dominant throughout the reaction. Consequently, small-sized crystals were formed without intercrystalline gaps, resulting in the formation of continuous crystal films.

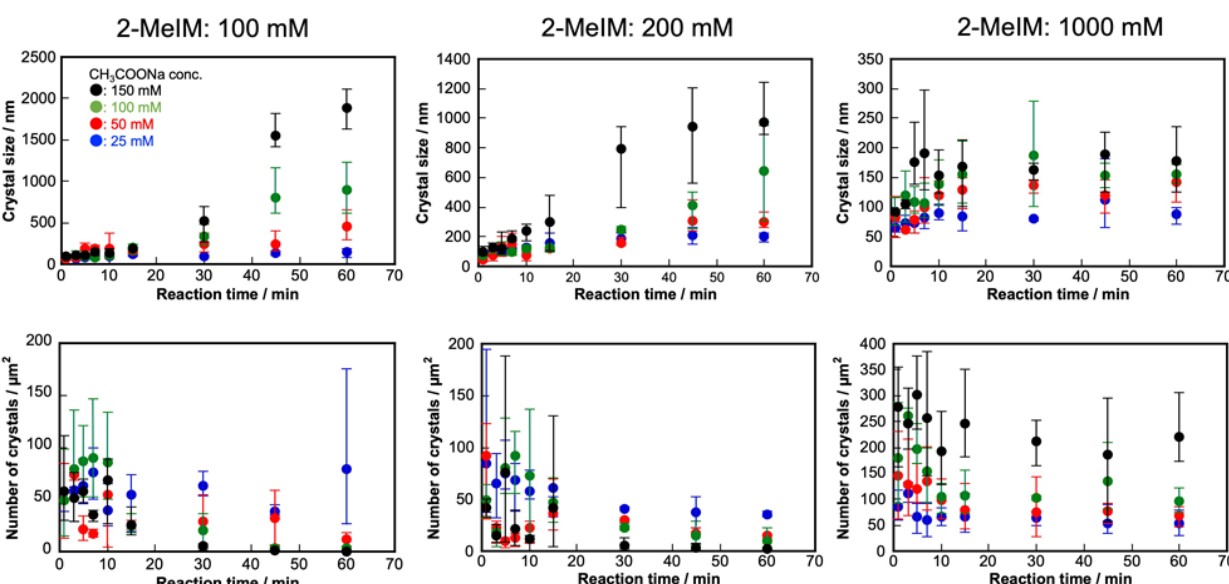

**Figure 4.** Changes in the size and number of the crystals prepared using reaction solutions containing different concentrations of 2-MeIM and $CH_3COONa$.

Based on these results, we proposed a formation mechanism in continuous ZIF-8 crystal films using the present approach (Figure 5). Firstly, the formation process can be classified by ligand concentration. When both ligand and additive concentrations are lower, the crystal growth is favored, leading to the formation of a small number of crystals with larger sizes. Next, as only the concentration of the additive increased, the nucleation showed a preferential approach in the initial stage of the reaction; then, the crystal growth was favored, resulting in the formation of continuous MOF crystal films. Finally, in the case of a higher ligand concentration, nucleation was the dominant process during the reaction, regardless of the additive concentration. Therefore, small-sized crystals were densely formed without gaps in the initial stage of the reaction.

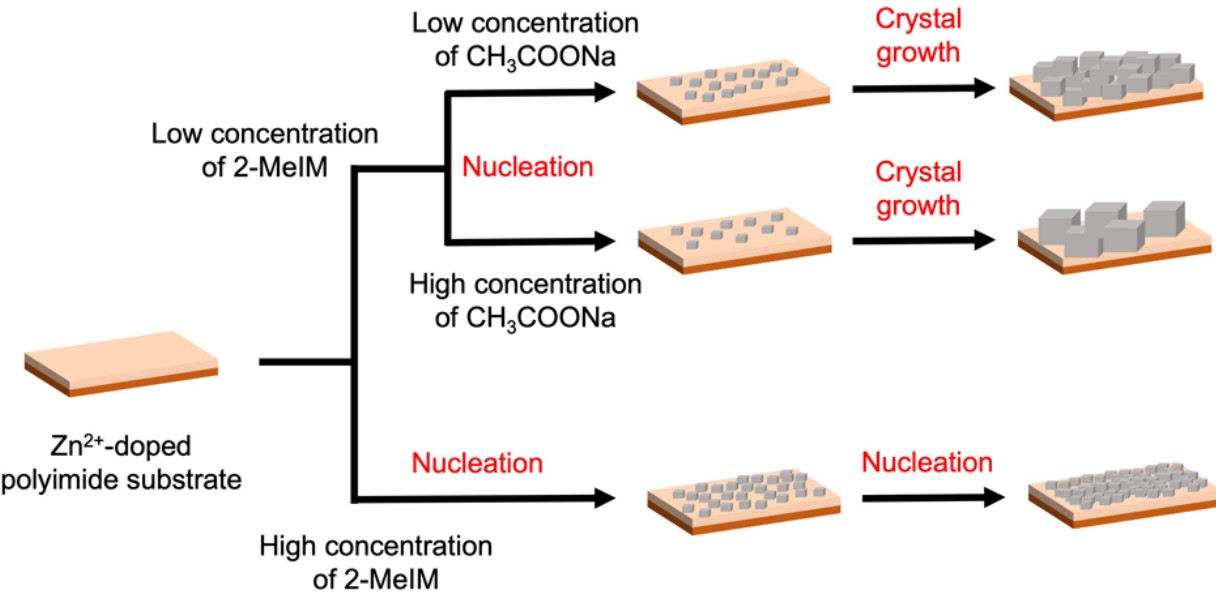

**Figure 5.** Schematic illustration of the proposed mechanism for the formation of ZIF-8 crystals using the present approach.

The effect of the methanol/$H_2O$ mixing ratio on the surface morphology of ZIF-8 crystal films: In the preparation of ZIF-8 crystals by conventional hydrothermal synthesis, the size of crystals depends on the solvent, with smaller-sized crystals forming in methanol solution and larger-sized crystals forming in aqueous solution. Therefore, the effect of the methanol/$H_2O$ mixing ratio on the surface morphology of the resulting samples was also investigated. To form a ZIF-8 crystal on the substrate, $Zn^{2+}$-doped polyimide films were immersed in a solution with various methanol/$H_2O$ mixing ratios containing 2-MeIM (100 mM) and $CH_3COONa$ (50 mM), followed by heating at 80 °C for 60 min. Surprisingly, ZIF-8 crystals were not formed on the substrate when the reaction was performed with only methanol solution. This may be due to the fact that the reaction temperature (boiling point: 64.7 °C) was too low to form frameworks. On the other hand, as the ratio of $H_2O$ in the reaction solution was increased, the number of crystals decreased and the size of crystals increased, indicating that higher content of $H_2O$ in the reaction solution led to a growth-rich process (Figure 6). XRD measurements revealed that the crystals formed in all samples were ZIF-8 frameworks. However, there were variations in the intensity of peak derived from the (110) plane of ZIF-8 framework. In the case of the higher content with methanol, the crystallinity of the resulting sample may be expected to be lower due to rapid nucleation. A decrease in peak intensity was also observed in the case of higher $H_2O$ content. This may be caused by the degradation of the -ZIF-8 crystals formed during the reaction. These results demonstrated that continuous ZIF-8 crystal films with controlled sizes can be formed by changing the ratio of methanol/$H_2O$ in the reaction solution.

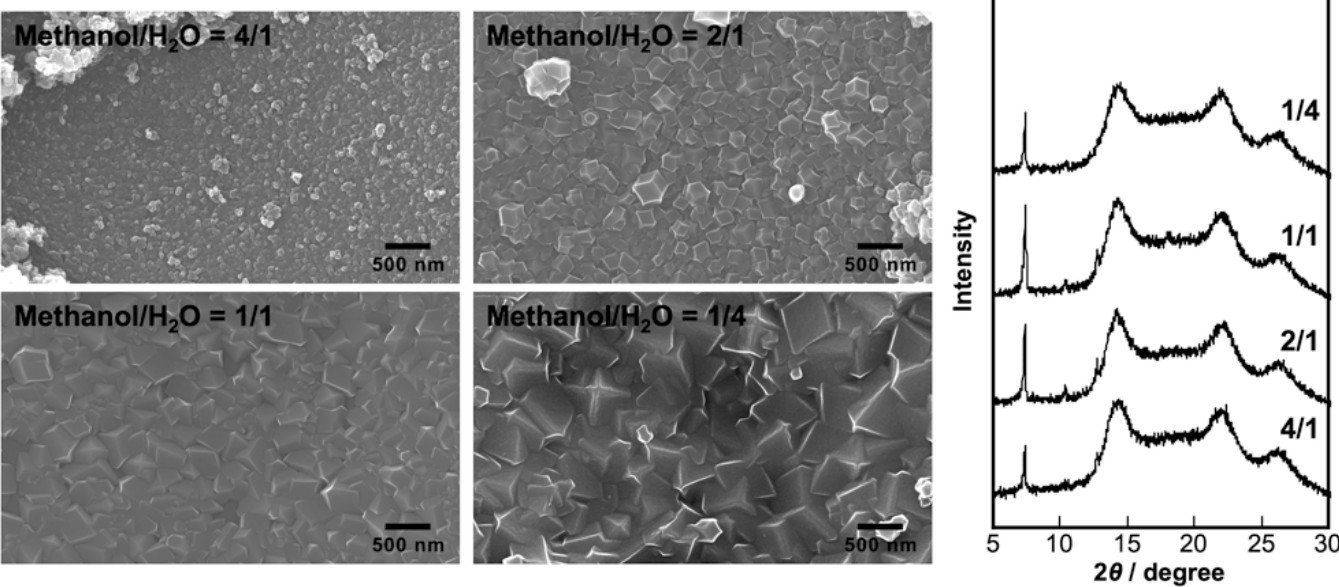

**Figure 6.** SEM images and XRD patterns of the samples prepared by using reaction solutions with different methanol/$H_2O$ ratio. (Concentration of 2-MeIM and $CH_3COONa$: 100 and 50 mM).

### 3. Experimental

Chemicals: Potassium hydroxide, zinc nitrate hexahydrate, sodium acetate, methanol, ethanol, and nitric acid were purchased from FUJIFILM Wako Pure Chemical Corp. 2-Methylimidazole (2-MeIM) was purchased from Tokyo Chemical Industry Co., Ltd. Pyromellitic dianhydride oxidianiline-type polyimide films (50 μm thick, Kapton 200H, Toray-Du Pont Co. Ltd.) were used as the polymer substrates. The films were cleaned via ultrasonication in ethanol at 25 °C for 5 min before use.

Preparation of ZIF-8 using a $Zn^{2+}$-doped polymer substrate: The polyimide films ($1 \times 2$ cm$^2$) were initially immersed in a 5 M aqueous KOH solution at 50 °C for 5 min followed by rinsing with large amounts of distilled water. The modified films were then immersed in a 50 mM aqueous $Zn(NO_3)_2$ solution at 25 °C for 20 min. After

rinsing with distilled water, the ion-doped polymer films were immersed in a solution (methanol/$H_2O$ = 1/2) containing 2-MeIM and $CH_3COONa$, followed by heating for 60 min at 80 °C with an aluminum heating block. The obtained samples were rinsed with the solution (methanol/$H_2O$ = 1/2) three times.

Characterization: The surface morphologies of the obtained films were observed via scanning electron microscopy (SEM; JSM-7001FA, JEOL). The average size of crystals was calculated by measuring the size at least 300 crystals using ImageJ software (version 1.53) (a public domain image processing and analysis program). The average number of crystals per μm$^2$ was calculated by measuring the number of crystals per 25 μm$^2$. X-ray diffraction (XRD) data were collected using a diffractometer (RINT-2200 Ultima IV, Rigaku) with Cu K$\alpha$ radiation. An inductively coupled plasma atomic emission spectroscopy (ICP AES; SPS 7800, Seiko Instruments) was utilized to quantify the amount of doped and eluted metal ions. The aqueous solution for measurements was prepared by immersing the resulting samples into 10% (*v/v*) nitric acid of aqueous solution. X-ray photoelectron spectroscopy (XPS; JPS-9010MC, JEOL) was utilized to characterization the chemical state of the samples. The obtained data were collected using the binding energy of the 4f orbital of Au.

## 4. Conclusions

In summary, we examined the impact of $CH_3COONa$ additive in the formation of ZIF-8 crystals on polyimide substrate doped with $Zn^{2+}$. The findings revealed that the use of $CH_3COONa$ prevented the leaching of unreacted $Zn^{2+}$ into the solution. Consequently, the reaction time was shortened from 72 to 1 h, and the volume of the reaction solution decreased from 1 L to 10 mL as compared to the use of conventional $NaNO_3$ additive. Furthermore, the detailed evaluation of the changes in the size and number of crystals during the reactions revealed that nucleation and growth rates can be controlled by adjusting the ligand and additive concentrations, and that continuous films consisting of ZIF-8 crystals can be formed by controlling these rates. Since this approach provides the formation of continuous MOF films on polymer substrates, it can be applied to the design and construction of MOF-based membranes for gas separation.

**Supplementary Materials:** The following supporting information can be downloaded at: https://www.mdpi.com/article/10.3390/inorganics12010021/s1, Figure S1. (A) XPS spectra of the obtained samples prepared by using reaction solution containing only CH3COONa, only 2-MeIM and both CH3COONa and 2-MeIM. (B) XPS spectra of the obtained samples prepared by different reaction time (concentration of 2-MeIM and CH3COONa: 100 and 50 mM). Figure S2. SEM images of the obtained samples prepared by using reaction solution containing only CH3COONa, only 2-MeIM and both CH3COONa and 2-MeIM. Figure S3. SEM images of the obtained samples prepared by using reaction solution containing concentrations of 2-MeIM (100 mM) and CH3COONa (25 mM). Figure S4. SEM images of the obtained samples prepared by using reaction solution containing concentrations of 2-MeIM (100 mM) and CH3COONa (50 mM). Figure S5. SEM images of the obtained samples prepared by using reaction solution containing concentrations of 2-MeIM (100 mM) and CH3COONa (100 mM). Figure S6. SEM images of the obtained samples prepared by using reaction solution containing concentrations of 2-MeIM (100 mM) and CH3COONa (150 mM). Figure S7. SEM images of the obtained samples prepared by using reaction solution containing concentrations of 2-MeIM (200 mM) and CH3COONa (25 mM). Figure S8. SEM images of the obtained samples prepared by using reaction solution containing concentrations of 2-MeIM (200 mM) and CH3COONa (50 mM). Figure S9. SEM images of the obtained samples prepared by using reaction solution containing concentrations of 2-MeIM (200 mM) and CH3COONa (100 mM). Figure S10. SEM images of the obtained samples prepared by using reaction solution containing concentrations of 2-MeIM (200 mM) and CH3COONa (150 mM). Figure S11. SEM images of the obtained samples prepared by using reaction solution containing concentrations of 2-MeIM (1000 mM) and CH3COONa (25 mM). Figure S12. SEM images of the obtained samples prepared by using reaction solution containing concentrations of 2-MeIM (1000 mM) and CH3COONa (50 mM). Figure S13. SEM images of the obtained samples prepared by using reaction solution containing concentrations of 2-MeIM (1000 mM) and

CH3COONa (100 mM). Figure S14. SEM images of the obtained samples prepared by using reaction solution containing concentrations of 2-MeIM (1000 mM) and CH3COONa (150 mM).

**Author Contributions:** K.A. (Kaito Araki), K.K. and T.T. performed the synthesis experiments and the characterization. T.T. conceived the experiments and supervised the project. T.T., Y.T. and K.A. (Kensuke Akamatsu) analyzed all date and discussed the formation mechanism of ZIF-8 crystals on the substrate. All authors have read and agreed to the published version of the manuscript.

**Funding:** This work was supported by JSPS KAKENHI Grant Number 21H05109 and 23K04897.

**Data Availability Statement:** The data presented in this study are available on request from the corresponding author.

**Conflicts of Interest:** The authors declare no conflict of interest.

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
