# Peer review of "Direct Formation of ZIF-8 Crystal Thin Films on the Surface of a Zinc Ion-Doped Polymer Substrate"

_inorganics, doi:10.3390/inorganics12010021_

Round 1

Reviewer 1 Report

Comments and Suggestions for Authors

This manuscript reports the impact of CH3COONa additive in the formation of ZIF-8 crystals on polyimide substrate doped with Zn2+. The results seem to be interesting and are suitable for publication in Inorganics. However, several points should be noticed and might be revised before the final acceptance and publication.

1. How do the authors demonstrate that the MOF membrane formed by this strategy is continuous and defect-free?

2. Electron microscope images of the membrane's cross-section are required to demonstrate the thickness of the membrane produced by this strategy.

3. Please describe and verify the interaction between the the polymer substrate and the MOF membrane. 

4. For the MOF-based membranes, one related publication can be refereed: Adv. Mater. 2016, 28, 2374.

Author Response

Comment 1 and 2:

  1. How do the authors demonstrate that the MOF membrane formed by this strategy is continuous and defect-free?
  2. Electron microscope images of the membrane's cross-section are required to demonstrate the thickness of the membrane produced by this strategy.

Response to comment 1:

  According to the reviewer’s suggestion, we have conducted the cross-sectional SEM observation of the samples obtained by using reaction solution containing organic ligand (100 mM) and CH3COONa (50 mM). These observations demonstrated that the NH2-MIL-53 (Al) crystals were densely packed in the films, which indicates that continuous thin films without voids and cracks could be successfully prepared by the approach. We have added the experimental data about the cross-sectional SEM images of the samples (Figure 1B) and these descriptions on page 2, line 85-87.

Comment 3:

Please describe and verify the interaction between the polymer substrate and the MOF membrane.

Response to comment 3:

As the reviewer pointed out, the interaction between MOF crystals and the substrate is very important. Therefore, we conducted a scotch tape test for the resultant MOF crystals on the polymer substrate, and fount that MOF crystals strongly attached on the polymer substrates (adhesive force > 1.18 N cm-1). We added this description on page 2, line 87-90.

Comment 4:

For the MOF-based membranes, one related publication can be refereed: Adv. Mater. 2016, 28, 2374.

Response to comment 4:

  According to the reviewer’s recommendation, we have added the reference (reference No. 23).

Reviewer 2 Report

Comments and Suggestions for Authors

Akamatsu et al. Present the preparation of a thin-film based on ZIF-8 crystals thought an ion-doped polymer substrate. Particularly the growth occurs thanks to an ion-exchange process between the additive and the Zn ions and the interfacial complexation with eluted zinc ions and 2 methylimidazole. Moreover, they show the study of the variation of the concentration of additive and the 2 methylimidazole. The results of these studies allow the control of the nucleation and growth rates, which gave rise to continuous films consisting of ZIF 8 crystals. The topic is not original because there are some other papers that show the preparation of thin-film based ZIF-8 on substrates. However, in this case, the authors have carried out a significant study. In this work, the study of the variation in the concentration apport and additional value to other similar works. The conclusions show the best results achieved by this work so they are consistent with what they have presented. In my opinion, the references are actual and appropriate.

The tables and figures are well-described and with good quality. Nevertheless, in the case of SEM images, I recommend changing the scale bar as well as the name inside the images to white instead of black.

Author Response

Comment:

The tables and figures are well-described and with good quality. Nevertheless, in the case of SEM images, I recommend changing the scale bar as well as the name inside the images to white instead of black.

Response to comment:

  According to the reviewer’s suggestion, we have changed to white instead of black. However, the scale bar and name inside the images were left unchanged in the original form (black) because they were easier to see.